# Sentiment analysis of medical record notes for lung cancer patients at the Department of Veterans Affairs

**Danne C. Elbers**[1,2,3]*, **Jennifer La**[2], **Joshua R. Minot**[1], **Robert Gramling**[4], **Mary T. Brophy**[2,5], **Nhan V. Do**[2,5], **Nathanael R. Fillmore**[2,3,6], **Peter S. Dodds**[1], **Christopher M. Danforth**[1]

**1** Vermont Complex Systems Center, University of Vermont, Burlington, VT, United States of America, **2** VHA Boston CSP Informatics, Department of Veterans Affairs, Boston, MA, United States of America, **3** Harvard Medical School, Boston, MA, United States of America, **4** Larner College of Medicine, University or Vermont, Burlington, VT, United States of America, **5** School of Medicine, Boston University, Boston, MA, United States of America, **6** Department of Medical Oncology, Dana-Farber Cancer Institute, Boston, MA, United States of America

* danne.elbers@va.gov

## Abstract

Natural language processing of medical records offers tremendous potential to improve the patient experience. Sentiment analysis of clinical notes has been performed with mixed results, often highlighting the issue that dictionary ratings are not domain specific. Here, for the first time, we re-calibrate the labMT sentiment dictionary on 3.5M clinical notes describing 10,000 patients diagnosed with lung cancer at the Department of Veterans Affairs. The sentiment score of notes was calculated for two years after date of diagnosis and evaluated against a lab test (platelet count) and a combination of data points (treatments). We found that the oncology specific labMT dictionary, after re-calibration for the clinical oncology domain, produces a promising signal in notes that can be detected based on a comparative analysis to the aforementioned parameters.

## Introduction

Estimating the sentiment expressed by text corpora has become popular in the social sciences, especially with the increasing prevalence of social media platforms such as Twitter and Facebook. The Hedonometer, developed by Dodds and Danforth [1, 2], estimates daily global happiness based on Twitter messages [3], and has been shown to reflect collective attention to global events. The instrument has been successfully utilized in a wide range of domains from quantifying happiness in green spaces [4] to identifying story arcs in palliative care conversations [5] and from understanding social amplification [6] to presidential engagement [7].

Here, for the first time, we explore the possibility of re-calibrating the Hedonometer sentiment scoring instrument to the clinical oncology domain and utilizing it to identify aspects of the patient trajectory through clinical notes. Understanding how care plans and communication are perceived by patients is crucial for continuous improvement of healthcare systems

**Data Availability Statement:** Patient related and note data cannot be shared publicly because it involves sensitive human subject data. Data may be available for researchers who meet the criteria

for access to confidential data after evaluation from VA Research and Development Committees. As a VA national legal policy (VHA Directive 1605.01), VA will only share patient data if there is a fully executed contract in place for the specific project. A common contractual mechanism utilized for this type of sharing is a 'Cooperative Research and Development' (CRADA) agreement. These contracts are typically negotiated in collaboration with VA national Office of General Council (OGC) and lawyers from the collaborating institution. These national sharing policies and standards also apply to deidentified data. In addition, if a contract is in place allowing sharing of deidentified data outside of VA, then VA national policy (VHA Directive 1605.01), states that deidentification certification needs to be met by Expert Determination. The expert determination requires independent assessment from an experienced master or PhD in biostatistics, from a third party not involved in the project, and may require outside funding to support. In addition, for an outside entity to preform research on VA patient data, IRB as well as VA Research and Development Committee approval is required for the specific project. Data requests may be sent to: VA Information Resource Center (VIReC) Building 18 Hines VA Hospital (151V) 5000 S. 5th Avenue Hines, IL 60141-3030 708-202-2413 708-202-2415 (fax) virec@va.gov.

**Funding:** The author(s) received no specific funding for this work.

**Competing interests:** The authors have declared that no competing interests exist.

and are currently often quantified only through narrative or survey based studies [8–10]. Sentiment scoring has been explored in the healthcare domain with mixed results. For example, Weissman et al. explored several sentiment scoring methodologies on notes recorded in the ICU [11] and made available through the MIMIC-III data set [12]. They found high variability between methods, but point out a strong association between sentiment and death, that had been seen previously [13]. Weissman et al. suggest that sentiment scoring methodology needs to be more strongly tailored to the healthcare domain and addressed with coverage of specific medical terminology. This argument was highlighted as well by McCoy et al. [14], who performed sentiment scoring on clinical notes utilizing a generic vocabulary scored for polarity (negative vs positive).

We agree that the sentiment associated with medical vocabulary differs greatly from its more common layperson usage. For example, the words 'positive' and 'negative' have a very different meaning in a clinical context then they do in generic use. A 'positive' clinical test often signifies an diagnostic indication of a health issue and is almost never a good event in medicine. In this manuscript we address this gap through execution of a data-driven approach to re-calibration of the original labMT sentiment dictionary. Secondly, we evaluate the domain-specific Hedonometer against several oncologic data points. Our goal is to gauge if the Hedonometer, after re-calibration, can detect a signal indicating clinical trajectory of cancer care using medical record notes.

## Materials and methods

### Selected data

The cohort used in this analysis consists of 10,000 randomly selected patients whose healthcare is managed by the Department of Veterans' Affairs (VA), and who were diagnosed with lung cancer between 2017 and 2019. A lung cancer diagnosis was determined based on the VA's Tumor Registry [15]. All notes (3,500,000+) were extracted from the VA's Corporate Data Warehouse (CDW) from date of diagnosis until 2 years after. To properly validate the signal strength of the re-calibrated Hedonometer instrument, no sub-selection was made based on note types, i.e. all notes were included.

This study was approved by the VA Boston Healthcare System Institutional Review Board (3209), under a waiver of informed consent. Access to raw data from the United States Department of Veterans Affairs' corporate data warehouse was granted by the VA Boston Healthcare System Institutional Review Board and the Veterans Health Administration National Data Systems.

### Re-calibration of the Hedonometer

First, we aimed to revise the LabMT word list for this clinical context, specifically so the sentiment scores were oriented towards the cancer domain. The original study produced the Language Assessment by Mechanical Turk sentiment ratings using an online survey on Amazon's crowdsourced work platform [2]. For the present study, we asked 5 health care providers (3 MDs, 2 nurses) with 10+ years experience in oncology to (re)assess 200 high-importance words (details on their selection will be provided later). In accordance with the original LabMT word list, words are scored on a scale from 1 – 9, with 9 being the happiest or most positive value and 1 being the least positive or most unhappy value. The instructions given to the SME's were the following:

'*The results of this survey will be used to measure the happiness of words in the context of lung oncology care notes. The overall aim is to asses how providers feel about individual words in their clinical context. Please rate the following individual words on a 9 point 'unhappy - happy' scale*

*with 5 being neutral. 1. Read the word. 2. Observe your emotional response. 3. Select score in accordance with your emotional response*'.

SME's were able to select a radio button response for each word, with a reminder of score meaning at 1 ('unhappy'), 5 ('neutral'), 9 ('happy').

After exclusion of stop words [16], the list of high-importance words for domain specific re-scoring was selected as follows. We aimed to use both (a) word frequency and (b) the likely difference between layperson and clinical context to identify labMT words most mismatched in sentiment. We chose 100,000 random notes from the larger dataset, and used these notes to design two distinct mechanisms for re-evaluation.

First, we counted all non-labMT words in the notes, and sorted them by frequency. The top 5 words in this lists were 'tab', 'medication', 'prn', 'reviewed' and 'provider'. Second, we parsed the notes for appearances of each anchor labMT word $w_i$, with labMT sentiment $h_i$, and identified the adjacent 5 words before and after in the notes. The average sentiment of these neighboring words across all notes was calculated to be $h_i^{amb}$, the so-called ambient sentiment, and compared with the original labMT rating $h_i$. Words $w_i$ were then sorted by the magnitude of $h_i^\delta = |h_i - h_i^{amb}|$ such that outliers for which the medical context sentiment $h_i^{amb}$ deviated substantially from the context independent sentiment $h_i$ could be identified. Words not scored in the original labMT study such as 'medication', found through frequency ranking, were artificially assigned an $h_i^\delta$ of 1. To combine these two word lists, we multiplied each word's frequency in the notes by $h_i^\delta$, and truncated after the top 200. The resulting 200 words were deemed most important to re-evaluate, given their prominence in the medical notes and their absent or poor context matching in labMT. Among the 200 words to be scored, a total of 66 new medical domain words were identified, and 134 labMT words were re-scored.

## Calculating sentiment

Sentiment for each note is calculated using the re-calibrated Hedonometer score list, which includes the original words, the original words with new scores, and the new words. In order to focus on the more informative sentiment contributions, words with values between 4 and 6 in the original word list [2] have been excluded from calculating a note's score. Due to the domain specific focus, words scored by the SME's have been included, regardless of score. A note's average score is subsequently calculated by, after excluding stop words [16], obtaining the frequency of the unique words occurring in the note, looking up their score in the re-calibrated word list, and multiplying the frequency of the word by its score. After which the sum is taken and the mean for the individual note is calculated [2], thus resulting in an average score for each note that can be used for further analysis. Words that are not scored are ignored in all calculations and in line with the Hedonometer strategy no further word cleaning, such as stemming, is performed [2].

## Comparative analysis

In order to validate the calibration of the Hedonometer instrument, several external parameters were selected to detect a signal in notes. Parameters that were selected for comparison included objective data in the form of lab test outcomes; platelet count, and a combination parameter based on medications, see Table 1. This specific set of parameters was chosen to assess the performance of the Hedonometer, since they are either very objective and often performed, such as the laboratory results known to be indicative of a patient's well being and cancer prognosis (platelet counts) [17–19], or a combination of medication and date associations that allow for comparison (treatments) [20–22]. In addition to validating the re-calibrated Hedonometer, the original Hedonometer was also evaluated for comparison.

**Table 1. Data collection.**

| Data Type | Start | End | Iteration |
|---|---|---|---|
| Notes | Date of Diagnosis | Date of Diagnosis + 24 months | all |
| Treatment | Start of Treatment | Start of Treatment + 6 weeks | daily |
| Platelet Count | Day of Test - 1 week | Day of Test + 1 week | all |

**Platelet count.** Platelet count was selected as an objective or hard measure, since it has been shown to be an indicator of how well a patient is feeling and has been linked extensively to cancer [17–19]. The hypothesis is that if a patient is not feeling well, which could relate to an abnormal platelet count, this 'unwell feeling' is recorded in clinical notes resulting in a drop in the sentiment score. Additionally, this lab test is ordered quite often, especially in cancer care, varying from every two weeks to daily depending on a patient's state. Since platelet count lab test outcomes come in a multitude of units, data was cleaned up and converted to one unit type ($10^9/L$) before inclusion. For this analysis, platelet count results were divided into three groups: low, normal and high. A low platelet count means below the threshold of 160 $10^9/L$, a normal platelet count is between 160 $10^9/L$ and 410 $10^9/L$, and a high platelet count above 410 $10^9/L$. All clinical notes one week before and one week after a platelet count test was performed were selected and the average score per note was calculated as described above. These notes were placed in the group in accordance with the lab test outcome. The three groups were subsequently tested for normality with Shapiro's as well as Anderson's tests. The outcome of these tests determines whether a one-way ANOVA or Kruskal-Wallis test is most appropriate to assess significance. If a significant difference between groups is found, a post-hoc test in the form of Tukey (parametric) or Conover (non-parametric) will be performed to find out which specific between-strata comparisons of platelet count appear to account for the overall difference in sentiment.

**Treatments.** As a combined data point to validate the re-calibrated Hedonometer against, cancer treatments were chosen. Specifically, chemotherapy, platinum (a form of chemotherapy), targeted and checkpoint therapies were evaluated. First, 39 medications were mapped to the treatments (see S3 Table), subsequently all these medications and the start of treatment were pulled from the Inpatient, Outpatient, Pharmacy and IV domains in VA's Corporate Data Warehouse (CDW). As treatments might be given in combination or sequence, patients can be counted multiple times. Notes were analyzed until six weeks after the start of the specific medication and associated with the treatment. The mean note score for each day was calculated, and a visual inspection of the mean note score fluctuation over time was provided and analyzed with SME's in clinical oncology. Additionally, word shift graphs [23] were created to gather understanding of which words influenced clear sentiment fluctuations. Lastly, the analysis was ran a second time, utilizing the LabMT dictionary only, to evaluate the difference with the re-calibrated dictionary.

## Software and systems

Cohort selection has been performed and data has been extracted from the VA's CDW using MSSQL. All subsequent calculations and analysis has been performed in Python 3.8, for analysis specifically the scipy, statsmodels and scikit packages have been used. Shared code can be found under the Code Availability section.

## Results

### Selected data

The selected cohort of 10,000 lung cancer patients resulted in a mean of 886 notes per patient per year and a standard deviation of 2180.

### Re-calibration of the Hedonometer

Please see supplemental information for a complete list of the 200 words selected based on their surrounding sentiment and frequency to be re-scored by SME's. Fig 1 shows that our selection of 200 words will cover 30% of note text.

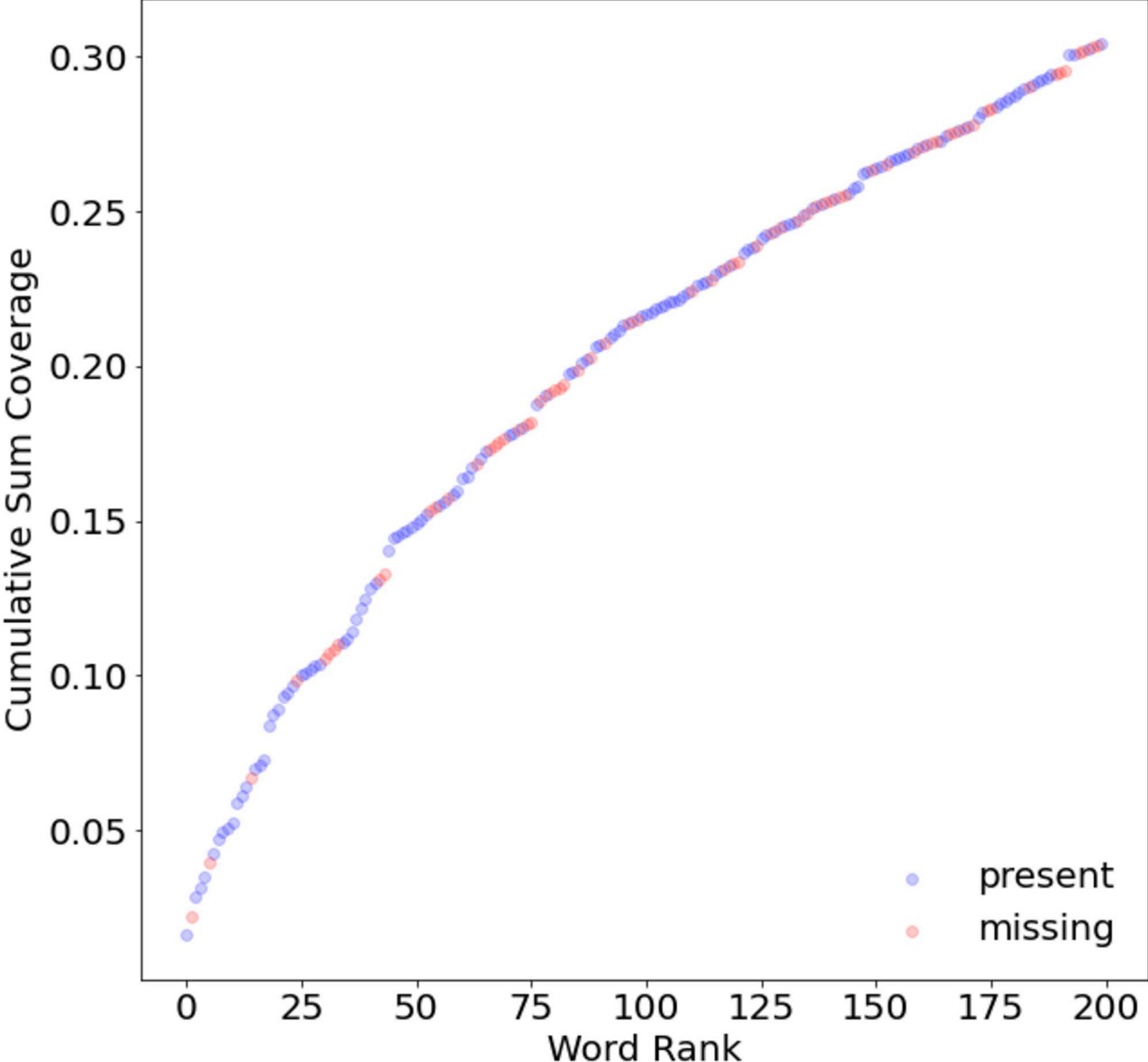

**Fig 1. Words are ranked based on the product of their clinical note coverage and the difference in word score to a word's ambient sentiment score.**

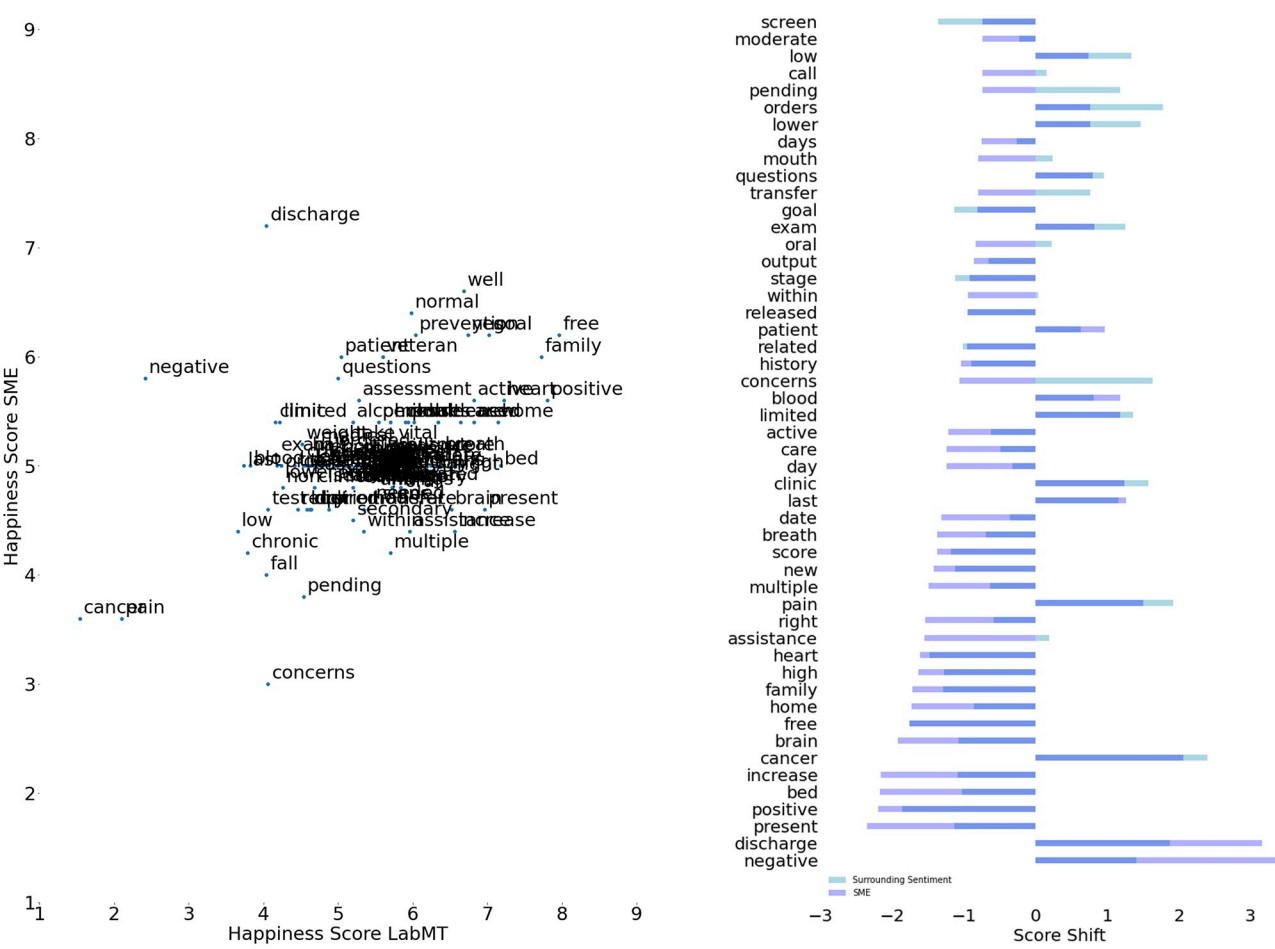

**Fig 2. Word score shifts due to calibration by SME's.** The figure on the left compares the LabMT scores to the scores assigned by the SME's. The right figure compares the ambient sentiment for each anchor term and the eventual assessment by the SME's.

Mean standard deviation of re-scoring the high-importance words between the SME's was 0.51 on the 9-point scale. The mean of re-scored words, 4.93, is lower then the mean of the original labMT dataset, namely 5.37. Full scores and standard deviation are available in S2 Table. The addition and updates of the oncology domain specific words result in a word list of 10,253, an increase of 66 words from the original 10,187. Examination of the re-scored words, see the left of Fig 2, shows that word 'positive' is scored less high by the SME's then in LabMT (5.6 vs. 7.8), while the word 'negative' is scored much higher (5.8 vs. 2.4). Other words standing out include 'discharge', 'pain', 'cancer', 'veteran' and 'family'. To compare, the right of Fig 2 shows the top 50 words with the largest shifts in word score for both the SME's evaluation and based on the surrounding sentiment calculation. A subset of random clinical notes was evaluated for confirmatory no's; standard questions generally answered with the word 'no'. Questions similar to 'have you recently travelled outside of the country?', with answers of 'no' were identified, however did not occur in high frequency. It appears that repetitive questions are often answered with the word 'negative', e.g. 'negative for fever, chills, blurring of vision, redness of eye, nausea' a word that was re-scaled by the SME's. The word 'no' itself is ranked 410, and was thus not highly influential on score calculation.

**Table 2. Platelet count — Post hoc conover test.**

|  | High | Low | Normal |
|---|---|---|---|
| High | 1 | 0.56 | $3.49 \times 10^{-06}$ |
| Low | 0.56 | 1 | $4.84 \times 10^{-05}$ |
| Normal | $3.49 \times 10^{-06}$ | $4.84 \times 10^{-05}$ | 1 |

## Comparative analysis

**Platelet count.** Every group with platelet count results, low, normal and high, was comprised evenly to contain 39882 unique notes. This number was equivalent to the group with the lowest count of notes (high group), for the larger groups a random subset was selected to create equal groups. All three groups failed both the Shapiro and Anderson normality tests. Although visual inspection of QQ-plots in addition to histograms did appear to come close to normality, the high number of samples might have played a factor in failing the tests. To be on the safe side, it was decided to test non-parametric, thus a Kruskal-Wallis was subsequently performed. The result of the Kruskal-Wallis test showed a significant difference ($p = 2.79 \times 10^{-06}$) and post-hoc Conover results are displayed in Table 2.

A significant difference in note score is present between the high platelet count group and the normal platelet count group, as well as between the low platelet count group and the normal platelet count group. However no difference was found between the high and low platelet count group. The high platelet count group had a median note score of 5.308; mean of 5.355, the low platelet count group a median note score of 5.316; mean of 5.356, while the normal platelet count group had a slightly higher median note score of 5.325; mean of 5.363. The original Hedonometer produced no significant result ($p = 0.62$) after the non-parametric test. This data similarly failed the Shapiro and Anderson normality tests.

**Treatments.** Evaluating the scoring of daily notes generated during the six weeks after the start of either one of the four different treatments; chemotherapy (633160 notes), platinum (317392 notes), checkpoint (230500 notes) and targeted (139908 notes), shows a cyclical weekly pattern, see Fig 3. Day 40 for targeted therapy has the lowest count of data points, namely 1980 notes.

To better identify what creates the dip in note scores on day 21, word shift plots are created, see Fig 4. Words related to lung cancer treatment such as 'lung', 'cancer', 'treatment', 'dose', 'mouth' and 'chemotherapy' appear to drive the score down. While more generic words, such as 'care', 'support', 'activity', 'patient' and 'independent', related to patient care seem to positively influence the note scores on peak days.

Rerunning the analysis with the original labMT dictionary showed a similar cyclical pattern, however the overall note scores were 0.3 points higher. This difference was attributed to the prevalence of missing clinical words in labMT, which were generally scored low by the SME's.

## Discussion

We have designed and implemented a data-driven method for re-calibration of an existing sentiment scoring instrument, the Hedonometer, in a specific healthcare domain. This re-calibration has been proposed numerous times in previous research on sentiment in clinical notes [11, 13, 14] as a solution to the mixed results found when utilizing existing vocabularies in this context. However, to the authors' awareness, re-calibration has not been done before. The re-calibration of the labMT sentiment dictionary for the oncology healthcare domain resulted in 200 re-scored words, which together cover 30% of clinical note text for the oncology domain,

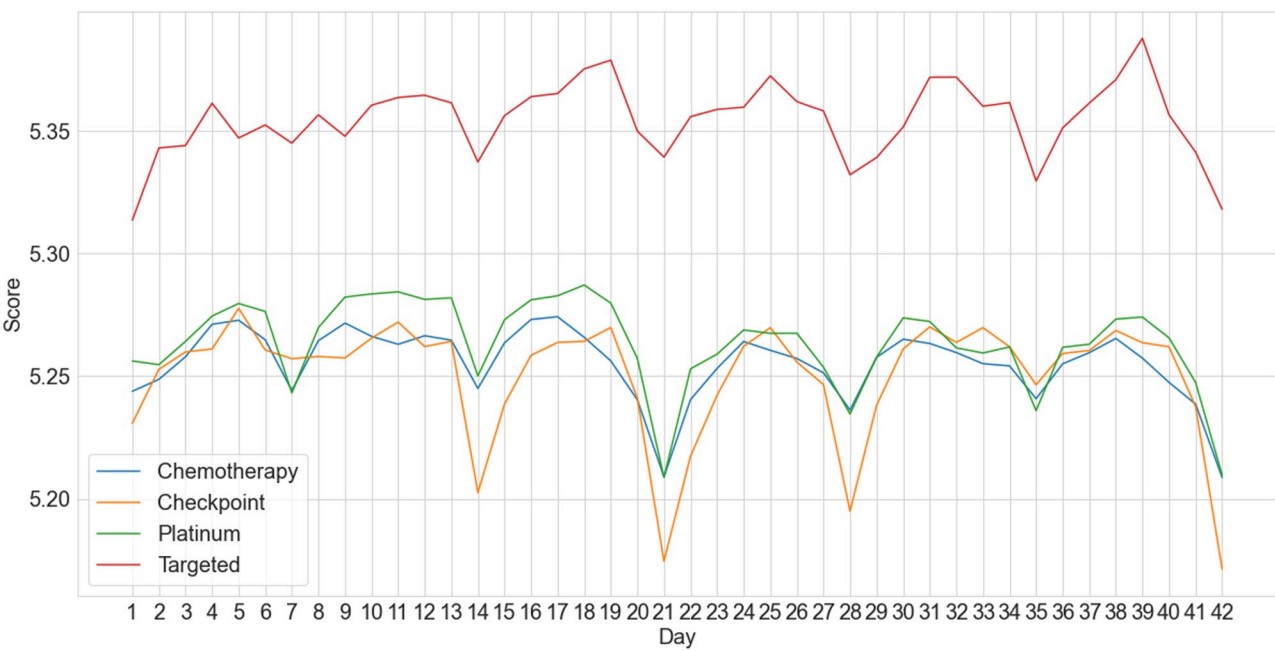

**Fig 3. Note scores on a day to day basis per treatment, starting at the day of treatment till six weeks from date of start of treatment.**

see Fig 1. Overall, this improves the coverage of words included in calculating the average score for a clinical note. A limitation of our re-calibration methodology is that only five SME's were included in the re-calibration, while a higher number would have improved its value. Identifying highly skilled SME's in the lung oncology domain with sufficient expertise is difficult and we are thankful for their given time.

The successful application of the domain-specific Hedonometer in our comparative analysis suggests that a signal is present and measurable in clinical notes. Validating the instrument on laboratory test outcomes, we found a significant difference in normal versus low or high platelet counts. This difference was not present when utilizing the original Hedonometer instrument. Additionally, a consistent cyclic effect is visible when scoring notes in relation to cancer treatment timelines, showing a comparable cycle for all treatments. This cycle can be explained by weekly provider visits discussing side effects and treatment itself, with a clear dip on day 21 for the chemotherapy (incl. platinum) and checkpoint treatments indicating the side-effects being at their worst. Targeted therapy tends to produce fewer side-effects and is often given to patients in better health, which could explain the better overall score and reduced fluctuation. Discussion with clinical SME's underscores this belief. When examining the score shift with day 21, treatment related words indeed appear to drive the note score down. Notably the text size is larger on day 21, associating more volume of note generation during visits. This negative shift is also consistent with Portier et al. [22], who perform sentiment analysis on online cancer support forums and find that side-effects of treatments are a topic with a clear negative sentiment shift. Though an explanation for the low dip in checkpoint treatment specifically was not presented by the SME's, it can be hypothesized that this treatment often includes further advanced cancer cases or cases in palliative care. It would be interesting to see if splitting out durvalumab, often given to less advanced cases, would generate higher sentiment. The outcome of the validation of the Hedonometer on these two measures does not preclude the sentiment changes from being related to other events in patients' care. Merely it shows presence of a measurable and explainable signal.

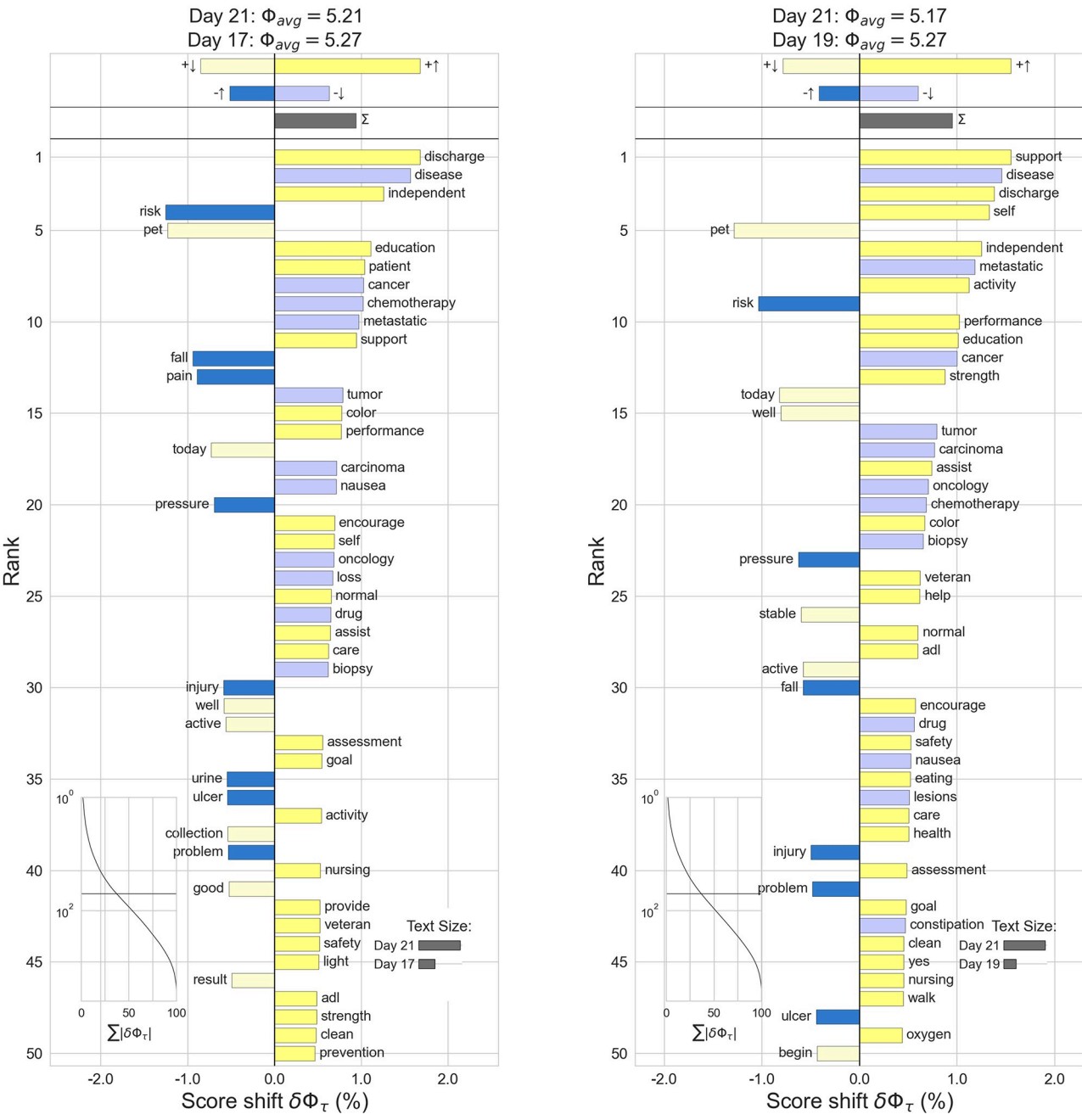

**Fig 4. Using notes authored on Day 21 of treatment as a reference, word-shift graphs detail the words influencing the drop in sentiment when compared with day 19 (left, Checkpoint Treatment) and and 17 (right, Chemotherapy Treatment).** Looking at the comparison between days 19 and 21 on the left, words appearing on the left side of the graph contribute positively to day 21, while words on the right side contribute positively to day 19 (there are many more of this type). For example, the relatively positive words 'support', 'discharge', and 'independent' are more common on day 19. The relatively negative words 'disease' and 'metastatic' are less common on day 19. Going against the overall trend, the relatively positive words 'today', 'well', and 'stable' are more common on day 21. The relatively negative words 'risk', 'pressure', and 'fall' are less common on day 21.

Our evaluation of the domain-specific Hedonometer is far from exhaustive, and future research should be done to understand better how and where the instrument can be utilized and what its limitations are in the oncology and healthcare domain. The finding that a signal appears present, detectable, and consistent is encouraging and the authors hypothesize next

research steps as stratification of the cohort and researching difference between note types, both of which have not been addressed here, and evaluation against other parameters. Other parameters could include, for example, the the VA - Frailty Index [24, 25], a well-validated performance measurement based on a large set of variables across different healthcare domains, or the VA-CAN score [26], surgery or neutrophil counts. If the signal remains robust, clinical and research implementation can vary from quality control of a new medication or treatment plan, to better understanding certain subgroups to quantifying differences between hospital, provider and nurse care. We hope that the domain-specific Hedonometer could provide informative value as a non-invasive, real-time lens into systems of care, making use of all notes entered into the Electronic Health Care Records (EHR). To foster future research in this area, the re-calibrated score list and associated code is made openly available along with this study.

## Supporting information

**S1 Table. Top 40 words based on rank of surrounding sentiment** * **text coverage.**
(CSV)

**S2 Table. Re-scoring outcomes from SME's.**
(CSV)

**S3 Table. Medication treatment matrix.**
(CSV)

## Acknowledgments

The authors like to thank Dr. Albert Lin, Dr. David Tuck, Karen Murray and Karen Visnaw for their help in re-calibration, and Dr. Albert Lin and Dr. Mikaela Fudolig for their input and suggestions. The views expressed are those of the authors and do not necessarily reflect the position or policy of the Department of Veterans Affairs or the U.S. government.

## Author Contributions

**Conceptualization:** Danne C. Elbers, Jennifer La, Mary T. Brophy, Nhan V. Do, Nathanael R. Fillmore, Christopher M. Danforth.

**Data curation:** Danne C. Elbers, Jennifer La.

**Formal analysis:** Danne C. Elbers.

**Investigation:** Danne C. Elbers.

**Methodology:** Danne C. Elbers, Jennifer La, Joshua R. Minot, Robert Gramling, Nathanael R. Fillmore, Christopher M. Danforth.

**Project administration:** Danne C. Elbers.

**Resources:** Danne C. Elbers.

**Software:** Danne C. Elbers.

**Supervision:** Danne C. Elbers, Robert Gramling, Mary T. Brophy, Nhan V. Do, Nathanael R. Fillmore, Peter S. Dodds, Christopher M. Danforth.

**Validation:** Danne C. Elbers, Mary T. Brophy.

**Visualization:** Danne C. Elbers, Joshua R. Minot.

**Writing – original draft:** Danne C. Elbers.

**Writing – review & editing:** Danne C. Elbers, Jennifer La, Joshua R. Minot, Robert Gramling, Mary T. Brophy, Nhan V. Do, Nathanael R. Fillmore, Peter S. Dodds, Christopher M. Danforth.

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
