## [Decision Letter · Decision Letter 0]

17 Aug 2022

PONE-D-22-11193Sentiment analysis of medical record notes for lung cancer patients at the Department of Veterans Affairs.PLOS ONE

Dear Dr. Elbers,

Thank you for submitting your manuscript to PLOS ONE. After careful consideration, we feel that it has merit but does not fully meet PLOS ONE’s publication criteria as it currently stands. Therefore, we invite you to submit a revised version of the manuscript that addresses the points raised during the review process. Your manuscript has been evaluated by one peer-reviewer and their comments are appended below.  The reviewer has raised a number of concerns that need your attention. They request additional information on methodological aspects of the study and the assessment of the results. In addition, the reviewer comments that the discussion section needs a deeper discussion of the limitations of the study. 

Could you please revise the manuscript to carefully address the concerns raised? Please note that we have only been able to secure a single reviewer to assess your manuscript. We are issuing a decision on your manuscript at this point to prevent further delays in the evaluation of your manuscript. Please be aware that the editor who handles your revised manuscript might find it necessary to invite additional reviewers to assess this work once the revised manuscript is submitted. However, we will aim to proceed on the basis of this single review if possible. 

We look forward to receiving your revised manuscript.

Kind regards,

Maria Elisabeth Johanna Zalm, Ph.D

Editorial Office

PLOS ONE

Journal Requirements:

2. Please include an ethics statement in your Methods section, clarifying whether or not the study was approved by an IRB, and provide the name of the IRB and any approval number.

4. Please provide additional details regarding participant consent. In the ethics statement in the Methods and online submission information, please ensure that you have specified (1) whether consent was informed and (2) what type you obtained (for instance, written or verbal, and if verbal, how it was documented and witnessed). If your study included minors, state whether you obtained consent from parents or guardians. If the need for consent was waived by the ethics committee, please include this information.

Reviewers' comments:

Reviewer's Responses to Questions

**Comments to the Author**

1. Is the manuscript technically sound, and do the data support the conclusions?

Reviewer #1: Partly

2. Has the statistical analysis been performed appropriately and rigorously? 

Reviewer #1: Yes

3. Have the authors made all data underlying the findings in their manuscript fully available?

Reviewer #1: No

4. Is the manuscript presented in an intelligible fashion and written in standard English?

Reviewer #1: Yes

5. Review Comments to the Author

Reviewer #1: This study re-calibrated the labMT sentiment dictionary on 3.5M clinical notes describing 10,000 patients diagnosed with lung cancer at the Department of Veterans Affairs. This is an interesting research topic in oncology domain, could be helpful for patient care in practice in the future. Some of my concerns focus on the following points:

1. Your goal is to gauge if the Hedonometer, after re-calibration, can detect a signal indicating clinical trajectory of cancer care using medical record notes. Have you compared with the original Hedonometer before re-calibration? Is the original Hedonometer capable of detecting such a signal? Then what the advantage of the Hedonometer after re-calibration?

2. You calculated the sentiment score of notes and evaluated against a lab test and treatment. Is this evaluation based on the assumption that the sentiment from notes is associated with lab tests or treatments? Would it be possible that the sentiment comes from other events? How did you deal with this possibility?

3. What is the rationale to determine the word frequency in your method: “obtaining the frequency of the unique words occurring in the note, looking up their score in the re-calibrated word list, and multiplying the frequency of the word by its score.”

4. In the sentence: “All clinical notes one week before and one week after a platelet count test was performed were selected and individually scored.” Who scored these clinical notes? What is the inter annotator agreement?

5. “According to SME’s in clinical lung cancer, this cyclical pattern can be explained due to patients’ weekly visits with their providers at which point they are asked about side-effects, bringing down the sentiment score of generated notes.” My question regarding this is if the dictionary before re-calibration can also produce the same result?

6. What do you mean here ? “The re-calibration of the labMT sentiment dictionary for the oncology healthcare domain resulted in 200 re-scored words, which together cover 30% of clinical note text for the oncology domain”

7. “Additionally, a consistent cyclic effect is visible when scoring notes in relation to cancer treatment timelines, showing a comparable cycle for all treatments.” Again, what is the result from the dictionary before re-calibration?

8. What is your limitation in this study?

English errors:

As treatments might be given in combination or sequence, patients can be counted more then once.

6. PLOS authors have the option to publish the peer review history of their article (what does this mean?). If published, this will include your full peer review and any attached files.

Reviewer #1: No

---

## [Author Response · Author response to Decision Letter 0]

14 Oct 2022

This study re-calibrated the labMT sentiment dictionary on 3.5M clinical notes describing 10,000 patients diagnosed with lung cancer at the Department of Veterans Affairs. This is an interesting research topic in oncology domain, could be helpful for patient care in practice in the future. Some of my concerns focus on the following points:

1. Your goal is to gauge if the Hedonometer, after re-calibration, can detect a signal indicating clinical trajectory of cancer care using medical record notes. Have you compared with the original Hedonometer before re-calibration? Is the original Hedonometer capable of detecting such a signal? Then what the advantage of the Hedonometer after re-calibration?

We compared with the original Hedonometer before re-calibration and have included these results in the updated manuscript. 

Briefly, the original Hedonometer was not capable of detecting a signal in the platelet comparison (p = 0.62), in contrast to the re-calibrated version. The re-calibrated Hedonometer also performed better than the original Hedonometer in regard to treatment analysis, as detailed in our response to item 5 below. We have updated the methods, results, and discussion to add comparison to the original Hedonometer.

2. You calculated the sentiment score of notes and evaluated against a lab test and treatment. Is this evaluation based on the assumption that the sentiment from notes is associated with lab tests or treatments? Would it be possible that the sentiment comes from other events? How did you deal with this possibility?

Our rationale for selecting platelet count as an objective measure is that it has been shown to be an indicator of how well a patient is feeling and has been linked extensively to cancer. The hypothesis is that if a patient is not feeling well, which could relate to an abnormal platelet count, this ‘unwell feeling’ is recorded in clinical notes, resulting in a drop in the sentiment score. However, we do acknowledge that the outcome of the validation of the Hedonometer on these two measures does not preclude the sentiment changes from being related to other events in patients' care. It merely shows presence of a measurable signal. We have added these points to the methods (under Methods – Platelet Count) and discussion.

3. What is the rationale to determine the word frequency in your method: “obtaining the frequency of the unique words occurring in the note, looking up their score in the re-calibrated word list, and multiplying the frequency of the word by its score.”

The rationale behind word score frequency is utilizing it to calculate the average score of a single note, this score is then further utilized in the analysis. We’ve updated the methods section accordingly with this. 

In addition, we’ve made the code available for calculating note scores through Github, see: https://github.com/delbers/SentimentAnalysisOfMedicalRecordNotes/blob/main/Code/score_note.py.

4. In the sentence: “All clinical notes one week before and one week after a platelet count test was performed were selected and individually scored.” Who scored these clinical notes? What is the inter annotator agreement?

Average note scores are calculated utilizing the recalibrated Hedonometer Word Score methodology as described under 3. As this is an automated method, inter annotator agreement is not applicable. Though slightly different from IAA, but comparable, the standard deviation of individual re-scored words by the Subject Matter Experts is made available in Table S2. The calculation method has been more strongly referenced in the methods section. 

5. “According to SME’s in clinical lung cancer, this cyclical pattern can be explained due to patients’ weekly visits with their providers at which point they are asked about side-effects, bringing down the sentiment score of generated notes.” My question regarding this is if the dictionary before re-calibration can also produce the same result?

The cyclical pattern was visible as well before recalibration; however it is not possible to find out what the oncology and healthcare specific words were that attributed to the cyclical pattern as highlighted in Fig 4, as these are not part of the original dictionary. Moreover, the average score was 0.3 points higher. This difference was attributed to the prevalence of missing clinical words in labMT, which were generally scored low by the SME’s. This information is present in the results section. 

6. What do you mean here ? “The re-calibration of the labMT sentiment dictionary for the oncology healthcare domain resulted in 200 re-scored words, which together cover 30% of clinical note text for the oncology domain”

This references back to the first section of the Results, where we describe that the 200 words selected for inclusion in the oncology specific word dictionary, and scored by Subject Matter Experts, cover 30% of all words in the clinical notes in the lung oncology domain within the VA. Thus, if we had only used the original Hedonometer dictionary the coverage of words being used to calculate the average sentiment for a clinical note would have been (on average) 30% less. The discussion has been updated with this reference and with a reference to Fig 1.

7. “Additionally, a consistent cyclic effect is visible when scoring notes in relation to cancer treatment timelines, showing a comparable cycle for all treatments.” Again, what is the result from the dictionary before re-calibration?

Addressed in 5.

8. What is your limitation in this study?

A limitation of our re-calibration methodology is that only five SME's were included in the re-calibration, while a higher number would have improved its value. Identifying highly skilled SME's in the lung oncology domain with sufficient expertise is difficult and we are thankful for their given time. 

Additionally, the outcome of the validation of the Hedonometer on these two measures does not preclude the sentiment changes from being related to other events in patients' care. Merely it shows presence of a measurable signal.

We like to emphasize that this is an explorative manuscript and is by no means exhaustive, future research should be done to understand better how and where the instrument can be utilized and what its limitations are in the oncology and healthcare domain. The finding that a signal appears present, detectable, and consistent is encouraging. These points have been added to the discussion.

English errors:

As treatments might be given in combination or sequence, patients can be counted more then once.

We thank the reviewer for noticing this mistake in English grammar and have updated the sentence as follows:

‘As treatments might be given in combination or sequence, patients can be counted multiple times’

---

## [Decision Letter · Decision Letter 1]

28 Dec 2022

PONE-D-22-11193R1Sentiment analysis of medical record notes for lung cancer patients at the Department of Veterans Affairs.PLOS ONE

Dear Dr. Elbers,

Thank you for submitting your manuscript to PLOS ONE. After careful consideration, we feel that it has merit but does not fully meet PLOS ONE’s publication criteria as it currently stands. Therefore, we invite you to submit a revised version of the manuscript that addresses the points raised during the review process.

We look forward to receiving your revised manuscript.

Kind regards,

Nabil Elhadi Elsayed Ali Omar, PharmD.,BCOP.,PhD(C)

Academic Editor

PLOS ONE

Reviewers' comments:

Reviewer's Responses to Questions

**Comments to the Author**

1. If the authors have adequately addressed your comments raised in a previous round of review and you feel that this manuscript is now acceptable for publication, you may indicate that here to bypass the “Comments to the Author” section, enter your conflict of interest statement in the “Confidential to Editor” section, and submit your "Accept" recommendation.

Reviewer #1: All comments have been addressed

Reviewer #2: (No Response)

2. Is the manuscript technically sound, and do the data support the conclusions?

Reviewer #1: Yes

Reviewer #2: Yes

3. Has the statistical analysis been performed appropriately and rigorously? 

Reviewer #1: Yes

Reviewer #2: Yes

4. Have the authors made all data underlying the findings in their manuscript fully available?

Reviewer #1: Yes

Reviewer #2: Yes

5. Is the manuscript presented in an intelligible fashion and written in standard English?

Reviewer #1: Yes

Reviewer #2: Yes

6. Review Comments to the Author

Reviewer #1: (No Response)

Reviewer #2: Thank you for your efforts, please find below my comments for improvements and some suggestions

Title: looks good and reflects the methodology and aim of the research

Introduction:

Good explanation of sentiment analysis in a simple way

Part of sentiment analysis negotiation and that you agree with them better present in the discussions and limit the introduction for the currently available evidence and use of this type of analysis in healthcare systems. Also, you may add information about how the previous sentimental analysis helped and affected the healthcare systems. You have to use evidence with references.

You mentioned what you have done in the introduction, again this should be in the methodology and the discussion part. Try assessing where is the gap you aim to fill with your research and mention it in the introduction.

I would suggest that you paraphrase and clearly state your aim at the end of the introduction section.

Methods:

In the selected data section, I suggest you clearly mention your inclusion and exclusion criteria.

Have you obtained Ethics Review Board? Please mention this in the methodology

I suggest moving this “This resulted in a mean of 886 notes per patient per year and a standard 52

deviation of 2180.” To the results section.

Where is the reference for lines 58,59?

What systems, programs, or program languages have you used? Please mention that

Please justify and explain lines 96-99 in more detail.

What did you use to perform the statistical analysis?

Results:

Line 152 is missing table number

I suggest that explaining the meaning of the results like in line 192 to be moved to the discussion section.

Discussion

I appreciate that you compared your data to other studies

Please elaborate more on the explanation of the result’s meaning and how it can affect the practice. Also, you may mention some of the limitations of your study. Furthermore, clear mention of what further studies need to be done in the future would be important and would enrich your manuscript.

7. PLOS authors have the option to publish the peer review history of their article (what does this mean?). If published, this will include your full peer review and any attached files.

Reviewer #1: **Yes**

Reviewer #2: No

<quillbot-extension-portal></quillbot-extension-portal>

---

## [Author Response · Author response to Decision Letter 1]

10 Jan 2023

Response to Reviewers (PONE-D-22-11193)

Based on comments from the reviewers the manuscript ‘Sentiment analysis of medical record notes for lung cancer patients at the Department of Veterans Affairs.’ has been adjusted. Our response can be found below. 

Reviewer 1

All comments have been addressed, no additional comments.

Reviewer 2

Thank you for your efforts, please find below my comments for improvements and some suggestions

Title: looks good and reflects the methodology and aim of the research

Introduction:

Good explanation of sentiment analysis in a simple way

Part of sentiment analysis negotiation and that you agree with them better present in the discussions and limit the introduction for the currently available evidence and use of this type of analysis in healthcare systems. Also, you may add information about how the previous sentimental analysis helped and affected the healthcare systems. You have to use evidence with references.

You mentioned what you have done in the introduction, again this should be in the methodology and the discussion part. Try assessing where is the gap you aim to fill with your research and mention it in the introduction.

I would suggest that you paraphrase and clearly state your aim at the end of the introduction section.

We have restructured the paper, in particular the introduction and discussion, as suggested.

Methods:

In the selected data section, I suggest you clearly mention your inclusion and exclusion criteria.

This has been updated in the Methods – Selected Data section with the addition of the VA’s Tumor Registry codes used to determine lung cancer. A reference has been added as well. 

Have you obtained Ethics Review Board? Please mention this in the methodology

The Ethics Review Board (Institutional Review Board) is added including the number of the study, in accordance with PLOS ONE’s guidelines, under Methods – Selected Data.

I suggest moving this “This resulted in a mean of 886 notes per patient per year and a standard 52 deviation of 2180.” To the results section.

This has been moved to the Results

Where is the reference for lines 58,59?

The appropriate reference has been added.

What systems, programs, or program languages have you used? Please mention that

A new subheader ‘Software and Systems’ had been added to Methods. 

Please justify and explain lines 96-99 in more detail.

This is further expanded upon in the Methods – Calculating Sentiment paragraph and includes a reference to the general practice of the Hedonometer, referencing the descriptive methodology paper by Dodds et al. (2011). 

What did you use to perform the statistical analysis?

Python 3.8, specifically the scipy, statsmodels and scikit packages. This has been added to the Software and Systems paragraph. 

Results:

Line 152 is missing table number

The table number has been added. 

I suggest that explaining the meaning of the results like in line 192 to be moved to the discussion section.

This was moved to the Discussion. 

Discussion

I appreciate that you compared your data to other studies

Please elaborate more on the explanation of the result’s meaning and how it can affect the practice. Also, you may mention some of the limitations of your study. Furthermore, clear mention of what further studies need to be done in the future would be important and would enrich your manuscript.

This has been elaborated upon and limitations have been added into the Discussion.

---

## [Editor Report · Decision Letter 2]

12 Jan 2023

Sentiment analysis of medical record notes for lung cancer patients at the Department of Veterans Affairs.

PONE-D-22-11193R2

Dear Dr. Elbers,

We’re pleased to inform you that your manuscript has been judged scientifically suitable for publication and will be formally accepted for publication once it meets all outstanding technical requirements.

Kind regards,

Nabil Elhadi Elsayed Ali Omar, PharmD.,BCOP.,PhD(C)

Academic Editor

PLOS ONE

Additional Editor Comments (optional):

Reviewers' comments:

<quillbot-extension-portal></quillbot-extension-portal>

---

## [Editor Report · Acceptance letter]

13 Jan 2023

PONE-D-22-11193R2 

Sentiment analysis of medical record notes for lung cancer patients at the Department of Veterans Affairs. 

Dear Dr. Elbers:

I'm pleased to inform you that your manuscript has been deemed suitable for publication in PLOS ONE. Congratulations! Your manuscript is now with our production department. 

Kind regards, 

on behalf of

Dr. Nabil Elhadi Elsayed Ali Omar 

Academic Editor

PLOS ONE